# Minimally Invasive Autopsy Practice in COVID-19 Cases: Biosafety and Findings [note 1]

**DOI:** 10.3390/pathogens10040412

**Published:** 2021-04-01

**Authors:** Natalia Rakislova, Lorena Marimon, Mamudo R. Ismail, Carla Carrilho, Fabiola Fernandes, Melania Ferrando, Paola Castillo, Maria Teresa Rodrigo-Calvo, José Guerrero, Estrella Ortiz, Abel Muñoz-Beatove, Miguel J. Martinez, Juan Carlos Hurtado, Mireia Navarro, Quique Bassat, Maria Maixenchs, Vima Delgado, Edwin Wallong, Anna Aceituno, Jean Kim, Christina Paganelli, Norman J. Goco, Iban Aldecoa, Antonio Martinez-Pozo, Daniel Martinez, José Ramírez-Ruz, Gieri Cathomas, Myriam Haab, Clara Menéndez, Jaume Ordi

**Affiliations:** 1ISGlobal, Barcelona Institute for Global Health, Hospital Clínic, Universitat de Barcelona, 08036 Barcelona, Spain; natalia.rakislova@isglobal.org (N.R.); lorena.marimon@isglobal.org (L.M.); melania.ferrando@isglobal.org (M.F.); myoldi@clinic.cat (M.J.M.); jhurtado@clinic.cat (J.C.H.); quique.bassat@isglobal.org (Q.B.); maria.maixenchs@isglobal.org (M.M.); vima.delgado@isglobal.org (V.D.); clara.menendez@isglobal.org (C.M.); 2Department of Pathology, Hospital Clínic, Universitat de Barcelona, 08036 Barcelona, Spain; pcastill@clinic.cat (P.C.); mtrodrigo@clinic.cat (M.T.R.-C.); jaguerrero@clinic.cat (J.G.); eortiz@clinic.cat (E.O.); amunozb@clinic.cat (A.M.-B.); ialdecoa@clinic.cat (I.A.); antonmar@clinic.cat (A.M.-P.); dmartin1@clinic.cat (D.M.); jramirez@clinic.cat (J.R.-R.); 3Department of Pathology, Faculty of Medicine, Eduardo Mondlane University, Maputo 1653, Mozambique; mamudoismail@yahoo.com.br (M.R.I.); carrilhocarla@hotmail.com (C.C.); fernandesfabiolacouto@gmail.com (F.F.); 4Department of Pathology, Maputo Central Hospital, Maputo 1653, Mozambique; 5Department of Microbiology, Hospital Clínic, Universitat de Barcelona, 08036 Barcelona, Spain; minavarr@clinic.cat; 6Centro de Investigação em Saúde de Manhiça (CISM), Maputo 1929, Mozambique; 7ICREA, Pg. Lluís Companys 23, 08010 Barcelona, Spain; 8Pediatrics Department, Hospital Sant Joan de Déu, University of Barcelona, 08950 Barcelona, Spain; 9Consorcio de Investigación Biomédica en Red de Epidemiología y Salud Pública (CIBERESP), 28029 Madrid, Spain; 10Department of Pathology, Kenyatta National Hospital (KNH), Nairobi 20723-00202, Kenya; edwin.walong@uonbi.ac.ke; 11RTI International (Research Triangle Institute), Research Triangle Park, NC 12255, USA; aaceituno@rti.org (A.A.); jeankim@rti.org (J.K.); cpaganelli@rti.org (C.P.); ngoco@rti.org (N.J.G.); 12Neurological Tissue Bank of the Biobank-IDIBAPS, Hospital Clínic of Barcelona, 08036 Barcelona, Spain; 13Institute of Pathology, Cantonal Hospital Baselland, 4410 Liestal, Switzerland; gieri.cathomas@ksbl.ch; 14Department of Pathology, Saarland University Medical Center, 66421 Homburg/Saar, Germany; myriam.haab@uks.eu

**Keywords:** autopsy, minimally invasive autopsy, minimally invasive tissue sampling, MIA, MITS, postmortem, post-mortem biopsy, COVID-19, biosafety, RT-PCR, diffuse alveolar damage

## Abstract

Postmortem studies are crucial for providing insight into emergent diseases. However, a complete autopsy is frequently not feasible in highly transmissible diseases due to biohazard challenges. Minimally invasive autopsy (MIA) is a needle-based approach aimed at collecting samples of key organs without opening the body, which may be a valid alternative in these cases. We aimed to: (a) provide biosafety guidelines for conducting MIAs in COVID-19 cases, (b) compare the performance of MIA versus complete autopsy, and (c) evaluate the safety of the procedure. Between October and December 2020, MIAs were conducted in six deceased patients with PCR-confirmed COVID-19, in a basic autopsy room, with reinforced personal protective equipment. Samples from the lungs and key organs were successfully obtained in all cases. A complete autopsy was performed on the same body immediately after the MIA. The diagnoses of the MIA matched those of the complete autopsy. In four patients, COVID-19 was the main cause of death, being responsible for the different stages of diffuse alveolar damage. No COVID-19 infection was detected in the personnel performing the MIAs or complete autopsies. In conclusion, MIA might be a feasible, adequate and safe alternative for cause of death investigation in COVID-19 cases.

## 1. Introduction

Coronavirus disease (COVID-19) caused by the severe acute respiratory syndrome coronavirus 2 (SARS-CoV-2) emerged at the end of 2019 in the city of Wuhan, in the Chinese province of Hubei [1]. In 2020, the disease was declared by the World Health Organization (WHO) as a global pandemic. The transmission of SARS-CoV-2 occurs mainly through respiratory secretions, although it can also occur through accidental introduction of virus-contaminated particles present on the skin of the fingers into the oral or nasal mucosae [2]. Most affected individuals present mild disease or are asymptomatic. However, older adults, individuals with comorbidities, and people with immunosuppression conditions are more prone to developing severe disease [3]. 

Complete autopsy is the gold standard to study how diseases affect different organs and systems [4,5]. However, autopsy studies in COVID-19-related deaths are still scarce in comparison with the abundance of clinical and epidemiological studies [6]. The main reasons for this paucity of data include logistical and biohazard concerns. Certified Biosafety Level-3 (BSL-3) facilities or rooms with negative air pressure, as well as appropriate personal protective equipment (PPE) are commonly required to perform autopsy in COVID-19 cases [7,8,9,10]. These requirements are difficult to meet during the current pandemic [11], especially in low-resource settings. Thus, there is a need for an alternative, less invasive, and theoretically safer approach to conduct post-mortem examinations safely in these settings [12]. 

Minimally invasive autopsy (MIA), also known as minimally invasive tissue sampling [13,14], is a needle-based approach aimed at collecting samples of the main organs and fluids without opening the body. MIA has been validated as an alternative to conventional autopsy [15,16,17,18]. The procedure markedly reduces disfigurement of the body compared with complete autopsy, which can increase acceptability by families of deceased patients [19]. Specific MIA protocols adapted to different age groups have recently been developed for use in middle- and low-income countries [14,20], with the method being successfully used in Sub-Saharan Africa, South America and in South Asia [14,15,21]. Remarkably, the procedure has been safely conducted in many cases with evidence of highly transmissible underlying infections, such as tuberculosis [22], Nipah Virus infections in Bangladesh [23] and yellow fever in Brazil [24]. 

The recommended measures for COVID-19 autopsy are highly variable [7,8,10,25,26,27,28,29,30]. Several guidelines recommend at least a BSL-3 autopsy facility to perform a complete autopsy safely in COVID-19 cases [10,29], whereas other guidelines alternatively recommend a negative pressure room [31] or whole room ventilation [32] with proper air filtration. Notably, evidence on the safety of performing MIA in COVID-19 cases is limited. Indeed, most MIA studies on COVID-19 do not report the biosafety precautions undertaken during the procedure [33,34,35,36,37,38,39,40,41,42,43]. Recently, biosafety recommendations for performing ultrasound-guided MIAs have been published [44], but the reported conditions [44] included a negative air-pressure room which is frequently not available in middle and low-resource settings. Moreover, the requirement of using an ultrasound further complicates the post-mortem procedure in terms of equipment and personnel preparation, and consequently, is barely feasible in many settings. To our knowledge, specific recommendations for non-ultrasound guided MIA procedure in COVID-19 cases have not yet been developed. 

Herein, we describe our experience in performing a small number of non-ultrasound-guided MIAs in deceased patients with polymerase chain reaction (PCR)-confirmed COVID-19, with the aim of highlighting the biosafety requirements adapted to the infrastructure conditions in middle- and low-resource settings and to evaluate the effectiveness of these measures to guarantee the safety of the staff conducting the procedure. Finally, we compared the performance of MIA with complete autopsy carried out on the same body.

## 2. Methods

### 2.1. Study Setting

In this observational study, we included patients who died in the intensive care unit of the Hospital Clinic of Barcelona, Spain with PCR-confirmed COVID-19 from 22 October 2020, to 31 December 2020 in whom the autopsy was requested by the clinician to clarify the cause of death or to evaluate the impact. 

The study protocols, which included MIA and complete autopsy, were approved by the Institutional Review Board of the Hospital Clínic of Barcelona (Protocol code HCB.2020.0577; Approved 01/05/2020 and Protocol code HCB.2020.0825; Approved 01/07/2020). Oral informed consent to perform the procedures was given by the relatives of the deceased. Written consent of the legal representative could not be obtained due to visiting restrictions in the hospital and COVID-19 perimetral restrictions in Catalonia. 

### 2.2. Preparation for the MIA: PPE Donning and Personnel

Appropriate PPE was available for each person involved in the procedure. The equipment included a scrub suit, waterproof suit with hood, waterproof apron, hat to protect hair, mask, eye protection (goggles), and waterproof shoe covers. For masks, a filter facepiece FFP3 mask was used and was covered by a surgical mask. All the personnel involved wore long-sleeved double gloves (surgical gloves). Figure 1 shows the reinforced PPE used for conducting MIA during the COVID-19 pandemic. All steps were easily followed by the people involved in the MIAs, and all the PPE items were considered as comfortable to wear. Table 1 shows the list of basic and desirable PPE, autopsy room requirements and other recommendations for conducting MIA in COVID-19 cases.

The number of people performing MIA was reduced to the minimum necessary, and no additional people were allowed in the room. Three people participated in the MIA procedure: (1) pathologist in charge of sample collection (MIA specialist); (2) first assistant responsible for managing jars, tubes and MIA form (basic checklist for sample collection), and (3) second assistant responsible for helping with the movement and manipulation of the body. All three participating persons were healthy individuals, with no known comorbidities, under 60 years of age, and with lengthy experience in the MIA procedure

### 2.3. Preparation and Performance of the MIA Procedure

Table 2 summarizes the types and characteristics of the tools used for fluid and tissue collection during MIA.

The protocol was adapted to cover specific morbid conditions or to fulfill particular research interests. Figure 2 shows the standard sampling MIA protocol adapted to COVID-19 cases. 

The MIA procedures were conducted in a conventional autopsy room (MIA room). Before starting the procedure, we ensured that the MIA room was well-ventilated, clean and had proper lighting. All the necessary equipment was transferred to the MIA room prior to bringing in the body. Previously prepared COVID-MIA kit boxes [14] were used, which included three needles, pre-labelled formalin jars and cryovials, and a basic checklist form for sample collection. Cryovials for PCR were pre-filled with lysis buffer solution (ATL buffer, Qiagen, Hilden, Germany). 

Movement and handling of the body were reduced to a minimum. Transfer of the body was conducted while in a waterproof sanitary bag to avoid excessive body fluid leakage [27]. Following the standard MIA protocol, we disinfected the areas of the body to be punctured and performed an external examination. Then, MIA was conducted following the COVID-19 MIA protocol: naso-oropharyngeal swab sampling, collection of cerebrospinal fluid using a 20G needle, and punctures of the liver, lungs, heart, and brain (trans-nasal and occipital approach) using 14–16G automatic biopsy needles. Finally, the bone marrow was sampled using a trephine needle, and a stool sample was obtained with a rectal swab. All the samples were taken with slow, measured movements. The first assistant managed the jars and ensured correct execution of the procedure. After each set of organ punctures, the pathologist performing the punctures decontaminated the outer surgical gloves with a wipe saturated with 70% ethanol. No splash, accidental puncture or other biohazard incident was registered during the procedures. 

Figure 1 illustrates the MIA procedure in a PCR-confirmed COVID-19 case.

### 2.4. After the MIA: Removal of the PPE, Autopsy Room Cleaning and Follow-Up of the Personnel Involved

Before exiting the MIA room, all the personnel removed their outer gloves and the internal gloves were cleaned and scrubbed with an alcohol solution. Afterwards, each PPE item was removed in the following order: (1) removal of the goggles; (2) removal of apron; (3) removal of suit and shoe covers; (4) removal of the surgical cap; (5) removal of the surgical mask; (6) removal of the FFP3 mask; (7) removal of the inner gloves. Between each of the steps, the internal gloves were sanitized and scrubbed with alcohol.

After completion of the MIA procedure, the body was transferred to the BSL-3 autopsy room. The basic autopsy room was gently cleaned with abundant hypochlorite solution and ventilated. 

### 2.5. Complete Autopsy Procedure

Immediately after the MIA, complete autopsy was conducted by another pathologist and technician neither of whom were involved in the MIA. Briefly, all the thoraco-abdominal organs were eviscerated and dissected for detailed gross examination. A sample was obtained from the main organs (both lungs, liver, heart, kidneys, spleen, bone marrow) for histological and microbiological analyses. In addition, samples from the airway, lymph nodes, testicles/ovaries, adrenal gland, skeletal muscle and skin were also collected. Additional samples from all the above organs were obtained for biobanking purposes.

The PPE doffing was performed following the same steps undertaken by personnel involved in the MIA. After completion of the complete autopsy, the body was returned to the morgue, and the BSL-3 room was carefully cleaned with abundant hypochlorite solution.

### 2.6. SARS-CoV-2 Testing of the Personnel Involved in the MIA and the Complete Autopsy

A list of personnel involved in the MIA and complete autopsy was created, and they were instructed to self-monitor within 14 days after the procedure. During the three weeks after the autopsy, all the personnel on the list underwent weekly SARS-CoV-2 rapid antigen testing (Roche Diagnostics Deutschland GmbH, Mannheim, Germany). Tests were performed by expert staff using the long nasal swab provided in the antigen test kits as indicated in the user manual. 

### 2.7. Handling of the Specimens, Pathological and Microbiological Methods, and Attribution of Cause of Death 

The microbiological samples of the MIA and complete autopsy were stored in a refrigerator at −80 °C. The tissue samples for biobanking were flash frozen and stored at −80 C. The histological samples were fixed in formalin and processed the next day.

The histological evaluation included hematoxylin and eosin stain in all samples and histochemical and/or immunohistochemical stains whenever required to achieve the diagnosis of cause of death. The slides of the MIA and complete autopsy cases were evaluated by different pathologists (NR and IA, respectively), blind to the findings of the other procedure.

In all cases, the microbiological study comprised real-time reverse transcriptase PCR (RT-PCR) testing for COVID-19 in cerebrospinal fluid, liver, lungs, and the central nervous system in MIA samples. Briefly, nucleic acids were extracted with an automated platform (Magna Pure, Roche Diagnostics GmbH, Mannheim, Germany), and the SARS-CoV-2 E gene (LightMix E gene, Roche) was amplified and detected in a LightCycler 480 thermocycler (Roche). 

After completing all the analyses, a panel composed of a pathologist, a microbiologist, and a clinician with expertise in infectious diseases evaluated all the data of the MIA and the clinical records and assigned the underlying cause of death and the conditions contributing to death. The team attributing the cause of death based on MIA results was blind to the results of the complete autopsy. In parallel, another panel of experts not involved in the MIA evaluated all the data of the complete autopsy. All conditions involved in the chain of events leading to death were independently coded for the MIA and the complete autopsy following the International Classification of Diseases, 11h revision (ICD-11).

The complete autopsy was considered as the gold standard and MIA diagnoses were compared to those of the complete autopsy. Cases in which the MIA diagnoses (underlying and immediate causes of death) matched those of complete autopsy were deemed as correctly classified.

## 3. Results

### 3.1. Clinical Characteristics of the Deceased Patients

During the study period, six patients underwent coupled MIA and complete autopsy procedures. The mean age of the deceased patients was 76.5 years (range 66–85). Three deceased patients were male and three were female. All six patients had underlying conditions (most commonly systemic arterial hypertension and diabetes mellitus type II), and four had multiple co-morbidities. One patient had been treated with immunosuppressive medication before acquiring COVID-19. During admission all patients were diagnosed with COVID-19 by a PCR test of a naso-oropharyngeal swab. The mean time interval between the onset of symptoms and death was 8.4 days (range 4–17). Four patients developed severe respiratory distress. Five of the patients were treated with mechanical ventilation. The clinical characteristics of the deceased patients, symptoms and comorbidities are summarized in Table 3.

### 3.2. Histological and Microbiological Results of the MIA, Complete Autopsy Findings and Cause of Death Attribution

All the MIA samples were representative of the targeted tissue, except in one case, in which a sub-occipital brain sample was unsuccessful (subcutaneous tissue). Most of the MIA findings were restricted to the lungs. Three out of the six patients showed predominantly late stage diffuse alveolar damage with lymphocytic interstitial infiltrate, one of whom presented superimposed acute necrotizing pneumonia and microthrombi in alveolar capillaries. One patient showed diffuse alveolar damage in exudative stage. Hyaline membranes were also observed in these four cases, albeit focally. In one case, acute necrotizing pneumonia was identified, with no evidence of diffuse alveolar damage. Finally, in the sixth case, we observed cores of cardiac tissue with acute myocardial infarction and destruction of the architecture. Moreover, the tissue cores with epicardial surface showed extensive hemorrhage and underlying tissue breaches, suggestive of wall rupture. Mild interstitial mononuclear infiltrate was focally observed in both lungs. 

All the lung and naso-oropharyngeal samples tested positive for SARS-CoV-2 (100%). The virus was identified in rectal swabs in four cases (66%). Brain tissue was COVID-19-positive in one case (17%). The liver and cerebrospinal fluid samples were negative in all cases. 

In all cases the complete autopsy confirmed the findings of the MIA. Table 3 summarizes the main histological findings in the lungs and other organs sampled during the MIA. 

An underlying, intermediate, and immediate cause of death were each established in the MIA and complete autopsy in all six cases. In all cases the assigned three diagnoses were the same for both methods. Both methods identified COVID-19 infection as an underlying condition in four cases (66%), directly leading to the pneumonia and acute respiratory distress syndrome causing death. In the other two cases (33%), both MIA and complete autopsy diagnosed COVID-19 as a condition contributing to death, but not directly involved in the chain of events leading to adverse outcome. Table 4 outlines the main pathological findings and the causes of deaths established using MIA for the six cases enrolled in the study. Figure 3 shows the most relevant histological findings identified in the MIA samples.

### 3.3. Follow-Up of the Personnel Involved in the MIA and Complete Autopsy

All the personnel involved in the MIA and the complete autopsy repeatedly tested negative for SARS-CoV-2 Rapid Antigen Test. None reported any symptomatology suspicious of COVID-19.

## 4. Discussion

In the present study we report our experience and preliminary results on the performance and biosafety of the MIA procedure in a small number of COVID-19 cases. To our knowledge, this is the first MIA study in COVID-19 that also includes complete diagnostic autopsy performed on the same body. As in other MIAs conducted in adults in Mozambique [15] and Brazil [45], we successfully obtained the samples included in the study protocol. Contrary to several recent MIA studies in COVID-19 [35,44,46], our findings suggest that ultrasound or computed tomography guidance is not necessary to consistently and successfully obtain representative, high-quality samples [14,15,16,17].

Importantly, the findings of the MIA in COVID-19 deaths were comparable to those of the conventional autopsy performed on the same body. Indeed, the accumulated evidence shows that the performance of MIA in COVID-19 deaths [33,34,35,36,37,38,39,40,41,44,46] is almost identical to that of the complete autopsies [44,47,48,49,50,51,52,53,54,55,56]. In our study, four deaths were attributed to COVID-19, whereas two of the deaths were considered as caused by other conditions; in these two cases the SARS-CoV-2 infection was considered a comorbidity. It was of note that all cases were correctly classified by the MIA when compared to the gold-standard complete diagnostic autopsy classification. Histological findings associated with SARS-CoV-2 infection mainly included diffuse alveolar damage in different stages, as observed in four cases in this series in which COVID-19 was the cause of death. In contrast with most of the MIA series [36,38,46] and complete autopsy studies [48,56,57], we rarely identified vascular damage, although our sample size was small. Finally, in keeping with previous studies using conventional autopsy [47,58] or MIA [36,39,47], we did not identify any lesions associated with SARS-CoV-2 in heart tissue. In one of our cases acute myocardial infarction was identified with both the MIA and the complete autopsy, but the condition was considered as not associated with COVID-19.

Similar to previously published data [36,38,44,46], the lung and naso-oropharyngeal samples consistently tested positive for COVID-19 PCR. Interestingly, all the liver samples were COVID-19 negative. Although other reports identified the virus in post-mortem liver samples [44,59], its presence was rarely confirmed by immunohistochemistry and the microbiological results were not associated with viral-like features or other histological findings [44,59]. In keeping with other reports, COVID-19 positivity in one of the brain samples was not correlated with any histological abnormalities [48]. Indeed, previous studies have shown that two COVID-19 receptors, TMPRSS2 and ACE2, are expressed at relatively low levels in the brain tissue [60]. Remarkably, all cases tested positive for SARS-CoV-2 in the nasopharyngeal swab, which contrasts with the 0% positivity rate observed in a recent Italian study [61]. This much higher rate could be partially due to the higher sensitivity of the test used in our study compared with that used in the aforementioned study (98.6% vs. 77%) [62]. Finally, we identified high percentages of COVID-19 positivity in stool samples, similarly to previous clinical studies [63,64].

In terms of biosafety, our experience indicates that all steps were easily followed, and all the PPE items were considered acceptable to wear by the personnel involved in the study. Although we used coverall suits, waterproof long-sleeved gowns might be a valid alternative for the procedure. Contrarily, our results suggest that full body suits that include powered air-purifying respirators are not necessary for MIA in cases with COVID-19. Similarly, although FFP3/N99 masks were used in this study, FFP2/N95 masks could probably be considered as sufficient when FFP3 are not available due to the limited production of aerosols during the MIA procedure. We safely used a basic autopsy room without negative-pressure to perform the MIAs. Indeed, the negative results in COVID-19 antigen tests among the personnel involved suggest that the generation of infective aerosols may be very limited during the MIA. Since COVID-19 testing is not universal in many countries and infected patients may not show typical signs of COVID-19 [65], it would be prudent to consider that any deceased person in an area with previous COVID-19 cases is potentially infected. Consequently, reinforced universal precautions should be taken for all the postmortem procedures conducted in any area during the COVID-19 pandemic.

Our study has several strengths and limitations. The strengths include the extensive previous experience of our group in conducting MIAs in Mozambique [15,18] and Brazil [45]. Contrary to several studies in which the sampling was restricted to the lungs [37,38,40,41,43], we successfully sampled a wide range of organs, similarly to the MIA procedures conducted in China [36], Brazil [46], Belgium [35] and United States [44]. The limitations include the small size of the study cohort and the lack of immunohistochemical or *in situ* hybridization staining for COVID-19 to confirm the presence of the virus in tissue. Secondly, our microbiological results do not include cycle threshold values, as the PCR assay was used for qualitative assessment (presence/absence).

In conclusion, the MIA approach offers an unparalleled opportunity to further study COVID-19 disease as MIA findings are almost identical to the findings of complete autopsy in COVID-19 cases. Although this MIA-complete autopsy series was relatively small, our preliminary results suggest that MIA findings in COVID-19 cases are comparable to those of complete autopsy and allow correct classification of deaths due to COVID-19 as well as deaths caused by other conditions in which SARS-CoV-2 infection is a contributing factor to death. The procedure can be safely conducted following a slightly adapted MIA protocol which includes reinforced personal protective equipment. In this regard, the absence of BSL-3 or negative-pressure autopsy rooms, which are rarely available even in high-income countries [11], should not be an obstacle for performing MIA to improve the scientific knowledge on this emergent disease. MIA can also be used in high-income settings to safely obtain tissue samples for research, whereas in low- and middle-income countries it might help to improve the COVID-19 death data.

## Figures and Tables

**Figure 1 pathogens-10-00412-f001:**
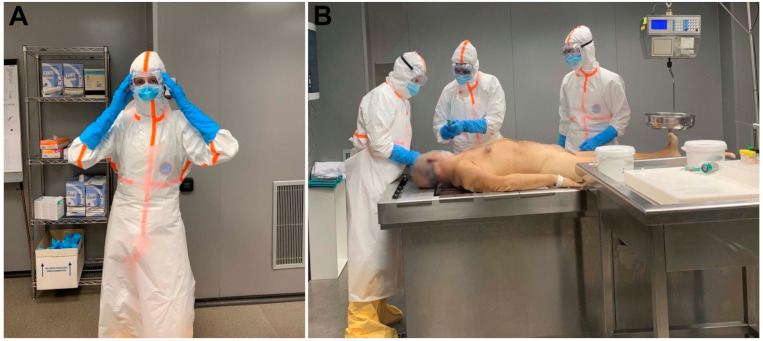
MIA preparation and sample collection. (**A**), Reinforced personal protective equipment is a fundamental part of minimally invasive autopsy (MIA) in COVID-19 cases. Waterproof suit with hood, waterproof apron, long-sleeved double gloves, waterproof shoe covers, FFP3 mask covered by surgical mask and eye protection (goggles). (**B**), MIA in a basic autopsy room; sample collection is performed by a pathologist (in the middle) with assistance of another person managing samplecollection tubes, jars, cryovials, and a third person helping to move the body. The autopsy room is clean and well-illuminated.

**Figure 2 pathogens-10-00412-f002:**
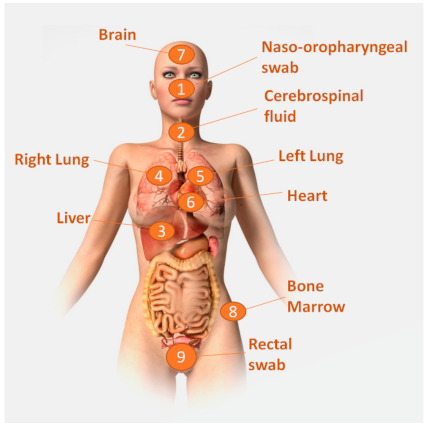
Minimally invasive autopsy (MIA) procedure adapted to possible or PCR-confirmed COVID-19 cases. This protocolized needle-based method consists of performing a series of punctures and collecting brush biopsies in an order shown with numbers, aimed at collecting highly informative fluids and tissues for pathological and microbiological analysis. (Illustration acquired from istockphoto.com; Photo ID: 136252326; Artist: leonello; the acquired image has been further modified).

**Figure 3 pathogens-10-00412-f003:**
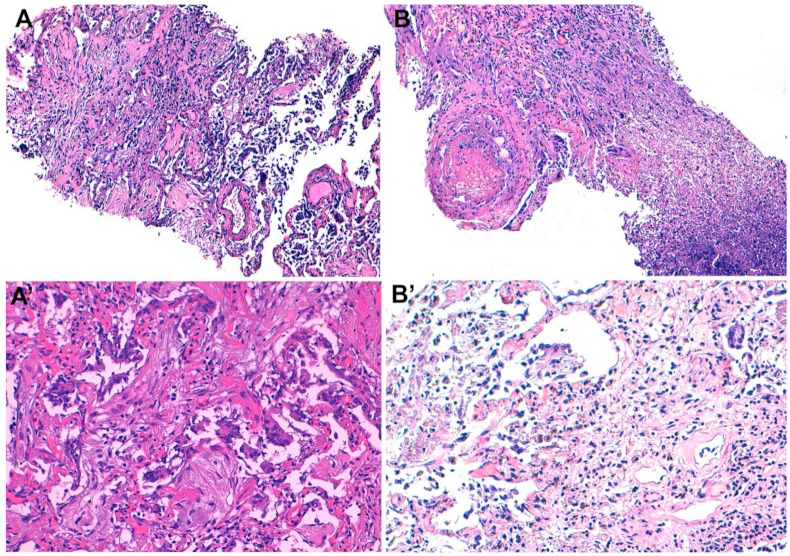
Histological samples obtained from the lungs during minimally invasive autopsy (MIA) from patients who died of coronavirus disease 19 (COVID-19). (**A**) diffuse alveolar damage in the proliferative stage with (**A’**) the presence of multinucleated giant cells, and hyperplasia of type II pneumocytes (**A**,**A’**): case 3, 76-year-old man). (**B**) features of diffuse alveolar damage with microthrombi and superimposed necrotizing pneumonia, and (**B’**) areas of fibrosis and lymphocytic inflammation (**B**,**B’**): case 4, 66-year-old woman).

**Table 1 pathogens-10-00412-t001:** List of basic requirements and other recommendations in terms of personal protective equipment and autopsy room characteristics for performing minimally invasive autopsy (MIA) in COVID-19 suspected or PCR-confirmed cases.

Personal Protective Equipment (PPE)			Autopsy Room		Other
Basic	Desirable		Basic	Desirable	Recommendations
Scrub suit	Plastic apron		Well-ventilated	Negative air pressure	All essential equipment prepared before the start of MIA (pre-labelling is desirable)
Waterproof long-sleeved gown	Waterproof gown covering the whole body orcoverall suit with hood	Clean	Clearly identified separation between “clean” and “dirty” areas	Use of pre-filled containers: Cryovials with lysis buffer for PCR samples Tubes with thioglycolate for cultures Jars with formalin for histological samples	
Boots	Specific closed shoes covered by waterproof shoe covers		Adequate lighting		Limit the number of personnel involved in the MIA and in the room
Long-sleeved double gloves for the MIA specialist (performing the punctures)	Long-sleeved double gloves for all the personnel involved in MIA				Thorough disinfection of the surfaces and tools during and after MIA
FFP3/N99	PAPR mask with a surgical mask		Free from unnecessary jars, tools and obstacles		Careful and slow doffing of PPE items
Lightweight safety goggles	Anti-fog lightweight safety googles or face shield				Provide a list of the personnel participating in MIA to the head of the department
Surgical cap to protect hair	Hood covering the surgical cap				

FFP: filter face protection; PAPR: powered air-purifying respirators; PCR: polymerase chain reaction.

**Table 2 pathogens-10-00412-t002:** Summary of tools and collection sites for the minimally invasive autopsy procedure.

Tissue/Body Fluid	Site of Collection	Mode of Collection	Needle/Tool	Type	Gauge	Needle Length (mm)
Naso-oropharyngeal secretions	Nasopharynx and oropharynx	Swabbing	Swab	Manual	N/A	N/A
Stool	Rectum	Brush sampling	Brush	Manual	N/A	N/A
Blood	Subclavian vein or heart	Needle aspiration	Quincke Spinal ^#^	Manual	20	100
Cerebrospinal fluid	Occipital approach to the cisterna magna	Needle aspiration	Quincke Spinal ^#^	Manual	20	100
Liver	Right lateral abdominal wall	Core needle biopsy	Monopty *	Automatic	14–16	115
Thorax (lungs/heart)	Right and left clavicular region down to the diaphragm for microbiology samples. Multiple random thoracic punctures for pathology.	Core needle biopsy	Monopty *	Automatic	14–16	100
Bone marrow	Anterior superior iliac crest	Trephine biopsy	T-Lok ™ trephine **	Manual	8	100
Central nervous system	Occipital approach, space between occipital bone and first vertebra. Trans-nasal approach through lamina cribrosa	Core needle biopsy	Monopty *	Automatic	16	200
Skin	Macroscopically detected lesions	Punch biopsy	Biopsy punch ^##^	Manual	5 mm	-

# Becton Dickinson, Franklin Lakes, NJ, USA, ## KAI Europe GMBH, Solingen, Germany, * BARD Biopsy Systems, Tempe, AZ; USA, ** Mana-Tech Ltd, Staffordshire, UK; N/A: not applicable.

**Table 3 pathogens-10-00412-t003:** Clinical characteristics of the six patients included in the study.

ID	Age/Sex	Comorbidities	Clinical Symptoms	Time from Symptoms Onset to Death (Days)
1	66/M	Diabetes mellitus type II, ischemic heart disease, kidney transplant, severe obesity,	Fever, dyspnea	4
2	88/F	Diabetes mellitus type II, cerebrovascular disease, chronic renal disease	Jaundice, cough, dyspnea	11
3	76/M	Hypertension	Fever, cough, hemoptysis, dyspnea	7
4	66/F	Chronic obstructive pulmonary disease, *cor pulmonale*, spondylitis	Dyspnea, mucopurulent sputum	17
5	78/M	Hypertension, diabetes mellitus type II, cerebrovascular disease	Lethargy	Unknown
6	85/F	Hypertension	Diarrhea	3

M: male; F: female.

**Table 4 pathogens-10-00412-t004:** Pathological findings, chain of events leading to death following the International Classification of Diseases, 11h revision (ICD-11), and the results of the reverse transcriptase PCR (RT-PCR) for COVID-19 in the minimally invasive autopsy (MIA) samples conducted in the six patients included in the study.

ID	Pathological Findings in the Lungs	Pathological Findings in Other Organs	Underlying Cause of Death	Intermediate Cause of Death	Immediate Cause of Death	NP/OP PCR	Lungs PCR	Liver PCR	Brain PCR	Stool PCR
1	Diffuse alveolar damage (exudative)	Extensive myocardial scarring	COVID-19 (test positive)	Pneumonia	ARDS	+	+	-	+	-
2	Diffuse alveolar damage (proliferative/exudative), focal bronchiolitis	Cholestatic hepatitis	COVID-19 (test positive)	Pneumonia	ARDS	+	+	-	-	+
3	Diffuse alveolar damage (proliferative/exudative)	-	COVID-19 (test positive)	Pneumonia	ARDS	+	+	-	-	+
4	Diffuse alveolar damage (proliferative), necrotizing pneumonia, microthrombi	Cholestasis	COVID-19 (test positive)	Pneumonia	ARDS	+	+	-	-	-
5	Diffuse necrotizing pneumonia	-	Aspiration pneumonia	Enterococcal septicemia	Septic shock	+	+	-	-	+
6	Mild interstitial infiltrates	Myocardial infarction with wall rupture	Acute myocardial infarction	Cardiac wall rupture	Cardiogenic shock	+	+	-	-	+

ARDS: acute respiratory distress syndrome; NP/OP: naso-oropharyngeal swab.

## Data Availability

All the data relevant to the study is shown in the manuscript and tables.

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
