# Peer review of "Minimally Invasive Autopsy Practice in COVID-19 Cases: Biosafety and Findings†"

_pathogens, 2021, doi:10.3390/pathogens10040412_

Round 1
Reviewer 1 Report
The manuscript has a good scientific level and describes a new method
in the evaluation of covid 19 disease. The study design is original andwell discussed. The bibliography appears detailed and very current.
I suggest improving the English language and grammar.
Author Response
Please find the point-by-point responses in the attached document
Reviewer 2 Report
Despite there are many other studies about MIA in COVID positive bodies, this paper could be a useful resource to people who are interested in MIA.
My only request is to add more specific information about the actual MIA procedure. I see that the authors summarize it in a figure and a table, but I would like to see a sort of step-by-step workflow that can be easily reproduced. Photos of the MIA kit and each step, would be a useful add to the manuscript.
The plagiarism score is 16% (excluding references). Please check if it is possible to rephrase some sentences to decrease it.
Thank you for your submission.
Author Response

(The authors gave the same response as above.)

Reviewer 3 Report
Thank you for letting me review this interesting paper: The topic is interesting and this study is coherent to the actual files of interest of the scientific community: the authors should check the followings items:
1) PPE should be written in the extended form when it is used for the first time;
2) the mythology of study is correct but they should clarify how the diagnosis is made by the pathologist (single pathologist for MIA and complete autopsy to evaluate the findings; double blind?);
3) the topic of the swabs should be clarified, and the authors should read and cite the following paper https://doi.org/10.3390/healthcare9020119 also in the evaluation of the causes of death;
4) references should be corrected according to the author guidelines of the journal. Finally, I suggest to accept the paper after this minor revision.
Author Response

(The authors gave the same response as above.)

Reviewer 4 Report
Dear authors, I carefully read your paper, and I find it scientifically well-sounding, results are well-presented, and the finding are interesting. The main limit is represented by the small number of patients included, but you correctly state it in the discussion.
I suggest to accept the paper, I just have few remarks:
- In the Table 1 you include among basic PPE a FFP2 mask. Considering that, even during a MIA, a lung puncture of lung is performed, this procedure may contribute to the presence of potentially contagous aeresol. For this reason, I think that, in this specific setting, a FFP3/N99 mask should be considered as basic PPE, while PAPR should be used as desirable item;
- In the same table, I noticed that you included FFP2 as basic and N95 as desirable. It is a mistake, given that the filtration of these masks is the same. They have different names because the certification office is different: FFP2/3 are certified by European Union, while N95 and N99 are certified by US-FDA. But the filtration capacity of FFP2 correspond to N95, while FFP3 is the same as N99. Please correct the table in order to be consistent;
- Given the small number of patient, you correctly said in your discussion that your result "suggest" some findings, except in one sentence, in which you write "we have demonstrated that the ultrasound or computed tomography guidance is not necessary". I think that you can not "demostrate" this with 6 patients only, please change to "Our findings suggest that...";
- Please consider to include among the reference: Fusco FM et al. A 2009 cross-sectional survey of procedures for post-mortem management of highly infectious disease patients in 48 isolation facilities in 16 countries: data from EuroNHID. Infection. 2016 Feb;44(1):57-64.It is an European-based survey and expert-opinion guidelines about post-mortem procedures in Highly Infectious Diseases, in which MIA was suggested as vialable option in these patients.
Author Response
Please find attached the point-by-point responses
